# Trends in cardiac rehabilitation rates among patients admitted for acute heart failure in Japan, 2009–2020

**Junghyun Kim** [1,2,3], **Jenny Jiang** [4], **Sophie Shen** [5], **Soko Setoguchi** [1,6]*

1 Institute for Health, Health Care Policy and Aging Research, Center for Pharmacoepidemiology and Treatment Science, Rutgers University, New Brunswick, New Jersey, United States of America, 2 Department of Preventive Medicine, Yonsei University College of Medicine, Seoul, Republic of Korea, 3 Institute for Innovation in Digital Healthcare, Yonsei University, Seoul, Republic of Korea, 4 Center for Observational Research and Data Science, Bristol Myers Squibb, Lawrenceville, New Jersey, United States of America, 5 Worldwide Patient Safety, Epidemiology, Bristol Myers Squibb, Princeton, New Jersey, United States of America, 6 Department of Medicine, Rutgers Robert Wood Johnson Medical School, Rutgers University, New Brunswick, New Jersey, United States of America

* soko.setoguchi@rutgers.edu

## Abstract

### Objectives

To describe inpatient and outpatient cardiac rehabilitation (CR) utilization patterns over time and by subgroups among patients admitted for acute heart failure (AHF) in Japan.

### Background

Cardiac rehabilitation (CR) is a crucial secondary prevention strategy for patients with heart failure. While the number of older patients with AHF continues to rise, trends in inpatient and outpatient CR participation following AHF in Japan have not been described to date.

### Methods

We conducted a retrospective cohort study of adult patients hospitalized for AHF in Japan between April 2008 and December 2020. Using data from the Medical Data Vision database, we measured trends in inpatient and outpatient CR participation following AHF. Descriptive analyses and summary statistics for AHF patients by CR participation status were reported.

### Results

The analytic cohort included 88,052 patients. Among these patients, 37,810 (42.9%) participated in inpatient and/or outpatient CR. Of those, 36,431 (96.4%) participated in inpatient CR only and 1,277 (3.4%) participated in both inpatient and outpatient CR. Rates of inpatient CR rose more than 6-fold over the study period, from 9% in 2009 to 55% in 2020, whereas rates of outpatient CR were consistently low.

**Data Availability Statement:** Soko Setoguchi and Junghyun Kim received consultancy fees from Bristol Myers Squibb. Jenny Jiang and Sophie Shen are employees of Bristol Myers Squibb, and Junghyun Kim was an employee of Rutgers

University when the research was conducted. The study was funded by Bristol Myers Squibb, NJ, USA (CV013-030). This does not alter our adherence to PLOS ONE policies on sharing data and materials. The datasets analyzed during the current study are not publicly available due to licensing agreement with Medical Data Vision Co., Ltd. MDV point of contact is CHANDARARITH LIEU (lieu_chandararith@mdv.co.jp) that other researchers would require to request access to the data.

**Funding:** Funding: The study was funded by Bristol Myers Squibb, NJ, USA (CV013-030). Disclosures: Soko Setoguchi and Junghyun Kim received consultancy fees from Bristol Myers Squibb. Jenny Jiang and Sophie Shen are employees of Bristol Myers Squibb, and Junghyun Kim was an employee of Rutgers University when the research was conducted.

**Competing interests:** Soko Setoguchi and Junghyun Kim received consultancy fees from Bristol Myers Squibb. Jenny Jiang and Sophie Shen are employees of Bristol Myers Squibb, and Junghyun Kim was an employee of Rutgers University when the research was conducted. The study was funded by Bristol Myers Squibb, NJ, USA (CV013-030). This does not alter our adherence to PLOS ONE policies on sharing data and materials. The datasets analyzed during the current study are not publicly available due to licensing agreement with Medical Data Vision Co., Ltd.

**Abbreviations:** AHF, acute heart failure; AF, atrial fibrillation; COPD, chronic obstructive pulmonary disease; CR, cardiac rehabilitation; DPC, diagnosis procedure combination; HF, heart failure; LOS, length of stay; MDV, Japan medical data vision database; NYHA, New York Heart Association; NOACs, novel oral anticoagulants.

## Conclusions

The rate of inpatient CR participation among AHF patients in Japan rose dramatically over a 12-year period, whereas outpatient CR following AHF was vastly underutilized. Further study is needed to assess the clinical effectiveness of inpatient CR and to create infrastructure and incentives to support and encourage outpatient CR.

## Introduction

Heart failure (HF) is a major global public health problem affecting 26 million people worldwide [1,2]. Acute HF (AHF) is a leading cause of mortality and unplanned hospital admission, especially in older populations [3–5]. Medical comorbidities precipitate rehospitalization and, when poorly managed, contribute to worsening of HF over time [6–8]. Psychosocial factors such as anxiety, depression, cognitive impairment and social isolation also confer increased risk of unplanned recurrent readmission or death following hospitalization for AHF [9–11]. Accordingly, comprehensive cardiovascular treatment is essential for successful management of AHF and reduction of associated disease burden.

Cardiac rehabilitation (CR) is a crucial secondary prevention strategy for patients with HF as part of a comprehensive intervention approach including exercise training and physical activity promotion, health education, cardiovascular risk management and psychological support, all personalized to individual needs [12]. CR is an effective intervention and is associated with diminishing morbidity and mortality [13–15]. The Japanese Circulation Society (JCS) Guidelines for Rehabilitation in Patients with Cardiovascular Disease recommend CR for AHF patients once their condition stabilizes [16]. Structured exercise training within a CR program is strongly recommended for this patient population [13,15].

In Japan, the overall incidence of HF was 0.8% in 2005 and is projected to increase rapidly as the population ages, reaching about 1.3 million by 2030 [17,18]. A 2007 nationwide questionnaire-based survey found that the rate of participation in outpatient CR following myocardial infarction was extremely low [19], and previous studies indicate that CR is underutilized in Japan [20–22]. However, to our knowledge, no study to date has described trends in rates of both inpatient and outpatient CR participation among AHF patients at the national level in Japan. To address this gap, we used data from a multi-institutional healthcare data set to describe inpatient and outpatient CR utilization patterns over time and by subgroups among patients admitted for AHF over a 12-year period in Japan.

## Methods

### Data source and study population

We conducted a retrospective cohort study using data from the Japan Medical Data Vision (MDV) database. The MDV includes diagnoses, procedures, pharmacy claims, and clinical information from Diagnosis Procedure Combination (DPC) which refers to the combination classification of diagnosis and procedure for a comprehensive evaluation of hospital care per patients including more than 40 million patients from over 460 hospitals. Our study cohort included all patients in the MDV aged ≥18 years with an initial inpatient admission for AHF between April 1st, 2009 and June 30th, 2020, who had at least one encounter recorded during prior 1-year baseline and post 6-month study period (April 1st, 2008 and December 31st, 2020). Hospitalizations due to AHF were identified based on a primary diagnosis of HF and

use of intravenous diuretics within 48 hours, in line with existing clinical trial endpoint definitions of AHF hospitalization [23,24]. Patients with acute coronary syndrome or any diagnosis of myocardial infarction during the AHF hospitalization were excluded.

## Cardiac rehabilitation (CR)

Inpatient CR generally comprises two stages in Japan. The first stage normally includes training on basic activities (e.g., sitting up in bed, sit-to-stand motions, walking), and the second stage includes progressive combined exercise training sessions (e.g., stretching, resistance, and aerobic training) as part of a 5-day-per-week program for patients with mild HF and a 3-day-per week program for those with severe HF. These regimens are based on the American College of Sports Medicine's guidelines for exercise testing and prescription and the Japanese Circulation Society's guidelines for rehabilitation of inpatients with cardiovascular disease [25,26]. For outpatient CR, the Japanese Circulation Society recommends that CR for patients with HF begin with supervised CR, followed by a combination of supervised and unsupervised CR 3 to 5 times per week for 5 months.

Inpatient CR was defined as participation in a CR program at least once during the hospitalization. Outpatient (post-discharge) CR was defined as CR participation within 90 days of discharge and prior to subsequent all-cause readmission. Duration of CR was defined as the number of days from CR initiation to conclusion. Time to initiation of inpatient CR was defined as the number of days from hospital admission to the CR start date, while time to outpatient CR initiation was defined as the number of days from hospital discharge to initiation of outpatient CR. The number of CR sessions was counted with one session equating to CR for 20 minutes, as per the Japanese reimbursement system.

## Covariates

Patient characteristics assessed included age at admission, sex, New York Heart Association (NYHA) classification at admission, smoking behavior, body mass index, comorbidities (coronary artery disease, diabetes, hypertension, stroke, atrial fibrillation (AF), anxiety, ventricular arrhythmia, pulmonary hypertension, peripheral vascular disease, COPD, chronic renal disease, valvular heart disease, hyperlipidemia, dementia; Charlson comorbidity index), medication on admission and prescribed during hospitalization (angiotensin-converting enzyme inhibitor, antiplatelets, β-blocker, statin, angiotensin receptor blocker, carperitide, digoxin, entresto, heparin, inotropic agents, novel oral anticoagulants (NOACs), vasodilator, warfarin), admission source, and length of stay (LOS).

## Statistical analysis

The study population was divided into four mutually exclusive groups with respect to CR participation: no CR, inpatient CR only, outpatient CR only, and participation in both inpatient and outpatient CR. Descriptive analyses and summary statistics for baseline characteristics and hospital course were reported by CR participation status. Trends in rates of inpatient and outpatient CR participation were described in the overall study sample, and trends in inpatient CR participation were described in subgroups defined by sex, age (18–54, 55–64, 65–74, 75–84, ≥85), NYHA class, BMI (underweight, normal, overweight, obese), smoking, diabetes status, and AF. We also assessed characteristics of inpatient CR including days from admission to initiation of the CR program, total number of CR sessions, number of days of CR, number of days between the first and last CR sessions, and number of sessions per day during the hospitalization. These characteristics were examined in the overall inpatient CR population as well as by sex and age (<65 vs. ≥65). All data management and statistical analyses were conducted

using SAS version 9.2 (SAS Institute, North Carolina, USA). Study data were de-identified and comply with patient confidentiality requirements, exemption from ethics approval and waiver of informed consent granted, and the study was approved by the institutional review board at Rutgers University (Pro2019002578).

# Results

## Study cohort

We identified a total of 278,968 patients admitted with a diagnosis of AHF during the study period. Among these, 117,345 patients received IV diuretics within 48 hours of admission; of these, 93,290 patients had at least 12 months of continuous enrollment before the index hospitalization during the baseline period. We excluded 48 patients aged younger than 18 and 5,189 patients who had diagnosis of myocardial infarction, yielding a final analytic sample included 88,053 patients (Fig 1). Within this sample, 37,810 patients (42.9%) participated in inpatient and/or outpatient CR. Of these patients, 36,431 (96.4%) participated in inpatient CR only,

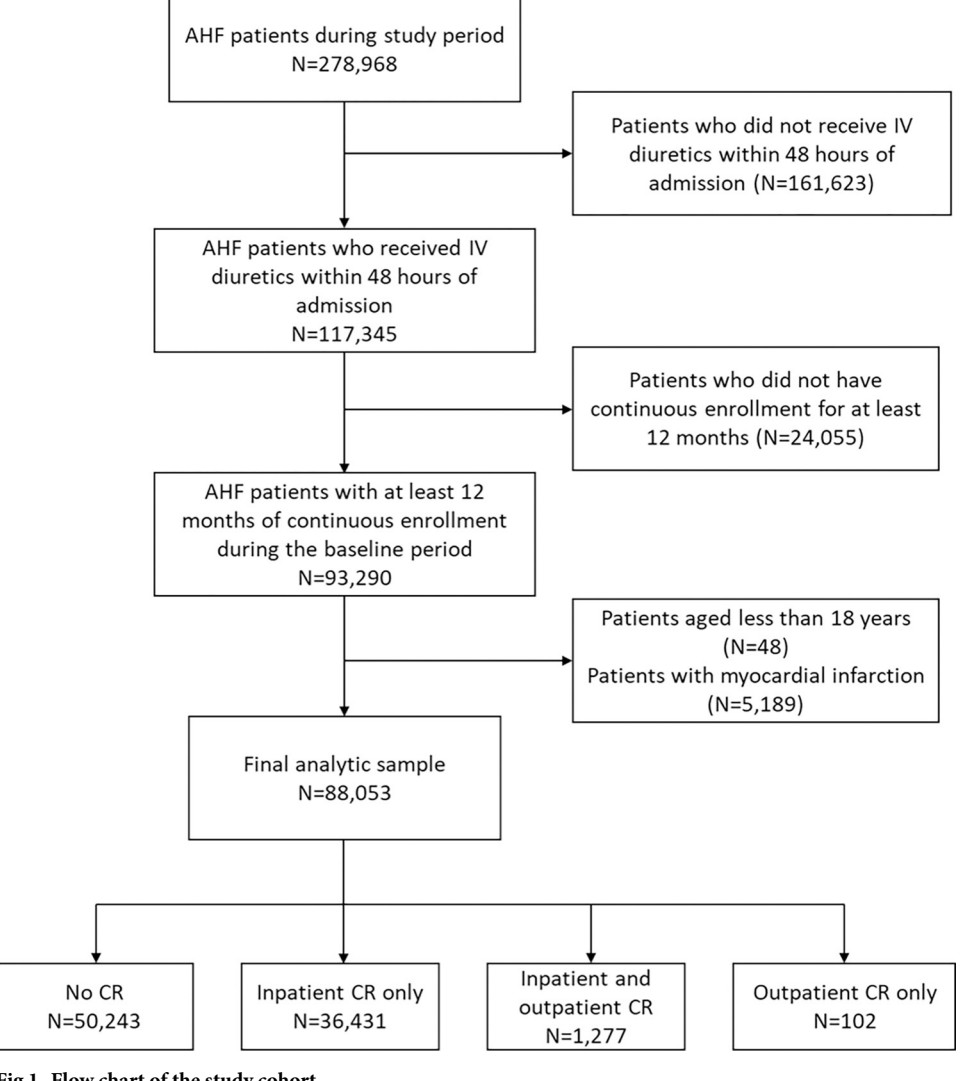

**Fig 1. Flow chart of the study cohort.**

1,277 (3.4%) participated in both inpatient and outpatient CR, and 102 (0.1%) participated in outpatient CR only.

## Baseline characteristics of AHF patients by CR participation

Baseline characteristics of the four CR groups are shown in Table 1. Patients who participated in any outpatient CR were substantially younger than those in the inpatient-only CR and no-CR groups, which exhibited similar age profiles to each other. Slightly fewer than half of patients in the inpatient-only CR group (45.9%) or the no-CR group (44.0%) were aged 85 or above, whereas fewer than 13% of patients who participated in outpatient CR were aged ≥85. Approximately two-thirds of patients in the outpatient CR groups were males, whereas roughly half of patients in the no-CR and inpatient-only CR groups were males.

Most patients in the study sample had a BMI within the normal range. Compared to patients who did not receive CR, those who participated in any CR had higher rates of several comorbidities including coronary artery disease, hypertension, AF, and hyperlipidemia. However, the prevalence of COPD and chronic renal disease were slightly higher in patients with no CR compared to those with any CR.

## Inpatient course of AHF patients by CR participation

Inpatient characteristics of the study population for the four CR groups are shown in Table 2. Among patients who participated in inpatient CR only, 15,460 (42%) were admitted through the ER. Patients with any inpatient CR were more frequently prescribed medications during hospitalization than those who did not receive any CR, with the exceptions of digoxin and entresto, for which usage did not greatly vary between the CR groups.

## Temporal trends in CR participation

Trends in inpatient and outpatient CR participation rates are shown in Fig 2. The rate of inpatient CR participation rose gradually, from 8.6% in 2009 to 54.9% in 2020, more than a 6-fold increase over this period. Rates of inpatient CR exhibited similar trends for males and females and for different age groups (Figs 3 and 4), though patients aged under 65 participated in inpatient CR slightly less frequently than older patients (S1A Fig in S1 File). Participation in inpatient CR declined over the study period among NYHA class 1 patients but rose among all other NYHA classifications (S1B Fig in S1 File). Patients with diabetes participated in inpatient CR slightly more frequently than those without diabetes throughout the study period (S1E Fig in S1 File), and patients with AF participated in inpatient CR more often than those without AF beginning in 2011 (S1F Fig in S1 File).

## Inpatient CR characteristics

Inpatient CR characteristics are shown in Table 3. Average time to initiation of inpatient CR following admission was 4.6 days. Inpatient CR patients received an average of 15.5 sessions over 12.4 CR days, with an average time from CR initiation to completion of 18.6 days. Inpatient CR characteristics did not vary greatly by sex or age. The mean LOS for patients with any inpatient CR was 27 days. No differences were observed by sex, but the mean LOS were slightly shorter for patients aged <65 (25 days) compared to those aged ≥65 (27 days). Mean LOS in the outpatient-only CR group was 9 days shorter than in the groups with any inpatient CR.

**Table 1. Baseline characteristics of the study population by CR participation.**

| | No CR | Inpatient CR only | Inpatient and outpatient CR | Outpatient CR only |
|---|---|---|---|---|
| | (N = 50,243) | (N = 36,431) | (N = 1,277) | (N = 102) |
| Age | | | | |
| Mean (SD) | 81.0 (11.0) | 81.8 (10.0) | 72.0 (11.8) | 73.8 (10.6) |
| Age category, N (%) | | | | |
| 18–54 | 1,466 (2.9) | 778 (2.1) | 118 (9.2) | 4 (3.9) |
| 55–64 | 2,541 (5.1) | 1,321 (3.6) | 152 (11.9) | 12 (11.8) |
| 65–74 | 7,216 (14.4) | 4,721 (13.0) | 391 (30.6) | 28 (27.5) |
| 75–84 | 16,930 (33.7) | 12,873 (35.3) | 467 (36.6) | 45 (44.1) |
| ≥85 | 22,090 (44.0) | 16,738 (45.9) | 149 (11.7) | 13 (12.8) |
| Sex | | | | |
| Female | 24,245 (48.3) | 18,052 (49.6) | 451 (35.3) | 35 (34.3) |
| Male | 25,998 (51.7) | 18,379 (50.5) | 826 (64.7) | 67 (65.7) |
| Smoking category | | | | |
| Nonsmoker | 31,214 (62.1) | 22,809 (62.6) | 666 (52.2) | 52 (51.0) |
| Smoker | 13,610 (27.1) | 9,849 (27.0) | 478 (37.4) | 36 (35.3) |
| Unknown | 5,419 (10.8) | 3,773 (10.4) | 133 (10.4) | 14 (13.7) |
| NYHA class | | | | |
| 1 | 1,495 (3.0) | 759 (2.1) | 38 (3.0) | 8 (7.8) |
| 2 | 5,819 (11.6) | 3,191 (8.8) | 117 (9.2) | 9 (8.8) |
| 3 | 8,381 (16.7) | 5,167 (14.2) | 161 (12.6) | 14 (13.7) |
| 4 | 8,870 (17.7) | 4,923 (13.5) | 170 (13.3) | 12 (11.8) |
| Unknown | 25,678 (51.1) | 22,391 (61.5) | 791 (61.9) | 59 (57.8) |
| BMI | | | | |
| <18.5 | 7,139 (14.2) | 5,323 (14.6) | 90 (7.1) | 55 (53.9) |
| 18.5–24.9 | 26,183 (52.1) | 20,156 (55.3) | 676 (52.9) | 27 (26.5) |
| 25–29.9 | 9,147 (18.1) | 6,722 (18.5) | 337 (26.4) | 7 (6.9) |
| ≥30 | 2,886 (5.7) | 2,052 (5.7) | 135 (10.6) | 9 (8.8) |
| Unknown | 4,888 (9.7) | 2,178 (6.0) | 39 (3.1) | 4 (3.9) |
| Comorbidities | | | | |
| Coronary artery disease | 6,869 (13.7) | 5,911 (16.2) | 261 (20.4) | 25 (24.5) |
| Diabetes | 15,792 (31.4) | 12,075 (33.1) | 474 (37.1) | 26 (25.5) |
| Hypertension | 34,028 (67.7) | 26,723 (73.4) | 988 (77.4) | 76 (74.5) |
| Stroke | 337 (0.7) | 204 (0.6) | 11 (0.9) | 1 (0.9) |
| Atrial fibrillation | 19,714 (39.2) | 16,479 (45.2) | 625 (48.9) | 44 (43.1) |
| Anxiety | 1,065 (2.1) | 754 (2.1) | 21 (1.6) | 2 (2.0) |
| Ventricular arrythmia | 1,886 (3.8) | 1,636 (4.5) | 114 (8.9) | 10 (9.8) |
| Pulmonary hypertension | 809 (1.6) | 557 (1.5) | 21 (1.6) | 1 (1.0) |
| Peripheral vascular disease | 5,018 (10.0) | 4,152 (11.4) | 151 (11.8) | 14 (13.7) |
| COPD | 7,983 (15.9) | 4,928 (13.5) | 163 (12.8) | 13 (12.8) |
| Chronic renal disease | 11,681 (23.3) | 7,875 (21.6) | 215 (16.8) | 17 (16.7) |
| Valvular heart disease | 9,638 (19.2) | 8,224 (22.6) | 253 (19.8) | 23 (22.6) |
| Hyperlipidemia | 14,133 (28.1) | 13,241 (36.4) | 620 (48.6) | 48 (47.1) |
| Dementia | 5,922 (11.8) | 4,049 (11.1) | 24 (1.9) | 1 (1.0) |
| Medication on admission | | | | |
| ACE inhibitors | 4,485 (8.9) | 3,007 (8.3) | 165 (12.9) | 14 (13.7) |
| Antiplatelets | 19,241 (38.3) | 13,058 (35.8) | 504 (39.5) | 41 (40.3) |
| β-blockers | 13,300 (25.5) | 9,700 (26.6) | 495 (38.8) | 41 (40.2) |

(*Continued*)

**Table 1.**  (Continued)

| | No CR | Inpatient CR only | Inpatient and outpatient CR | Outpatient CR only |
|---|---|---|---|---|
| | (N = 50,243) | (N = 36,431) | (N = 1,277) | (N = 102) |
| Statin | 10,007 (19.9) | 7,325 (20.1) | 362 (28.4) | 27 (26.5) |
| ARBs | 13,671 (27.2) | 8,295 (22.8) | 355 (27.8) | 33 (32.4) |
| Carperitide | 887 (1.8) | 740 (2.0) | 48 (3.8) | 4 (3.9) |
| Digoxin | 2,112 (4.2) | 1,054 (2.9) | 48 (3.8) | 8 (7.8) |
| Entresto | 1,917 (3.8) | 1,073 (3.0) | 48 (3.8) | 4 (3.9) |
| Heparin | 4,847 (9.7) | 4,020 (11.0) | 190 (14.9) | 16 (15.7) |
| Inotropic agents | 394 (0.8) | 441 (1.2) | 37 (2.9) | 2 (2.0) |
| NOAC | 5,354 (10.7) | 4,352 (12.0) | 205 (16.1) | 11 (10.8) |
| Warfarin | 7,145 (14.2) | 4,203 (11.5) | 216 (16.9) | 17 (16.7) |
| Vasodilator | 18,688 (37.2) | 17,063 (46.8) | 732 (57.3) | 57 (55.9) |

## Discussion

### Principal findings

We measured trends in participation rates for inpatient and outpatient CR in Japan between 2009 and 2020. Rates of inpatient CR rose dramatically over the study period, from 9% in 2009 to 55% in 2020. However, rates of outpatient CR remained steady at under 2% over the same period. Among patients who participated in inpatient CR, the mean durations from admission to initiation of CR and from initiation to completion were 4.6 and 18.6 days, respectively. These patients received an average of 15.5 sessions during hospitalization, with an average of 12.4 days spent on inpatient CR.

**Table 2.  Inpatient characteristics of the study population.**

| | No CR | Inpatient CR only | Inpatient and outpatient CR | Outpatient CR only |
|---|---|---|---|---|
| | (N = 50,243) | (N = 36,431) | (N = 1277) | (N = 102) |
| Admitted through ER | 18,121 (36.1) | 15,460 (42.4) | 365 (27.9) | 26 (25.5) |
| Admission source | | | | |
| From home | 35,371 (70.4) | 31,131 (85.5) | 1,177 (92.2) | 91 (89.2) |
| From nursing/welfare care facility | 4491 (8.9) | 2,517 (6.9) | 8 (0.6) | 1 (0.99) |
| From wards | 941 (1.9) | 809 (2.2) | 13 (1.0) | 1 (0.99) |
| Other/unknown | 9,440 (18.8) | 1,974 (5.4) | 79 (6.2) | 9 (8.8) |
| Medication given during hospitalization | | | | |
| ACE inhibitors | 9,609 (19.1) | 9,946 (27.3) | 496 (38.8) | 36 (35.3) |
| Antiplatelets | 25,089 (49.9) | 21,644 (59.4) | 760 (59.5) | 60 (58.8) |
| β-blockers | 24,784 (49.3) | 23,785 (65.3) | 1,047 (82.0) | 70 (68.6) |
| Statin | 12,511 (24.9) | 12,336 (33.9) | 551 (43.1) | 37 (36.3) |
| ARBs | 17,000 (33.8) | 13,730 (37.7) | 516 (40.4) | 42 (41.2) |
| Carperitide | 16,790 (33.4) | 14,480 (39.8) | 508 (39.8) | 36 (35.3) |
| Digoxin | 4,298 (8.6) | 2,741 (7.5) | 114 (8.9) | 7 (6.9) |
| Entresto | 1,906 (3.8) | 1,337 (3.7) | 57 (4.5) | 3 (2.9) |
| Heparin | 18,576 (37.0) | 17,447 (47.9) | 766 (60.0) | 53 (52.0) |
| Inotropic agents | 5,091 (10.1) | 5,170 (14.2) | 253 (19.8) | 17 (16.7) |
| NOAC | 11,018 (21.9) | 11,500 (31.6) | 461 (36.1) | 25 (24.5) |
| Vasodilator | 18,688 (37.2) | 17,063 (46.8) | 732 (57.3) | 57 (55.9) |
| Warfarin | 9,263 (18.4) | 7,081 (19.4) | 298 (23.3) | 26 (25.5) |

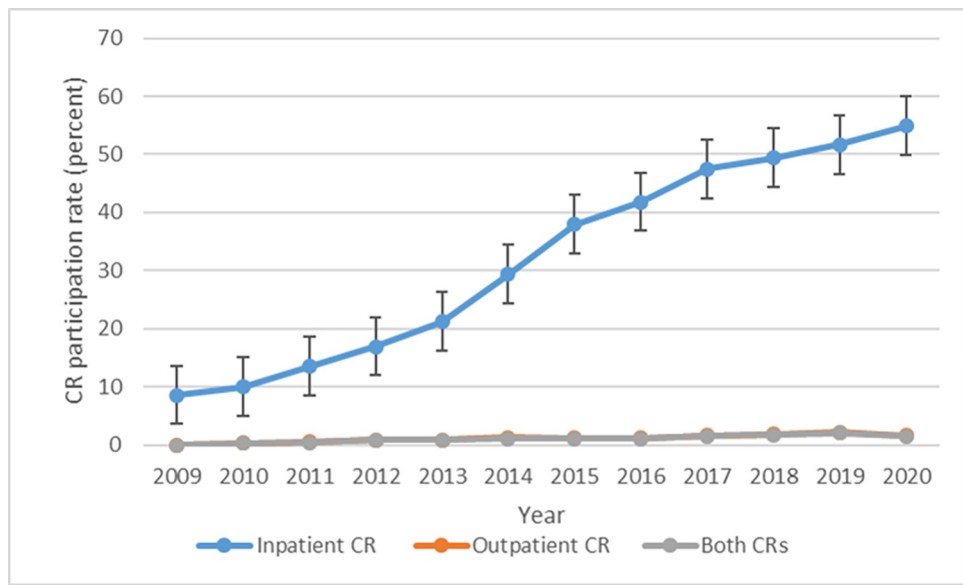

**Fig 2. Trends in inpatient and outpatient CR participation rates.**

## Results in the context of what is known

The prolonged length of inpatient care and high participation rate in an intensive inpatient CR program for AHF patients are unique practices in Japan and have been described in previous studies [27–30]. Our study builds on this work using more recent data and describing trends in both inpatient and outpatient CR in AHF patients. Our study confirms that LOS is long for AHF hospitalizations in Japan (mean 27 days) compared to Europe, where registry-based

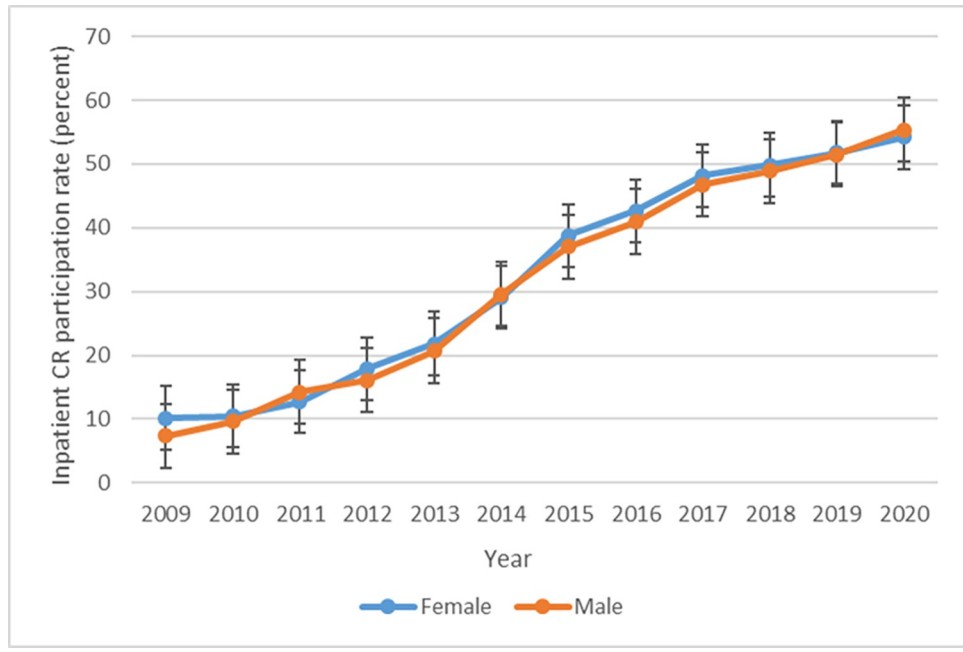

**Fig 3. Trends in inpatient CR participation rate by sex.**

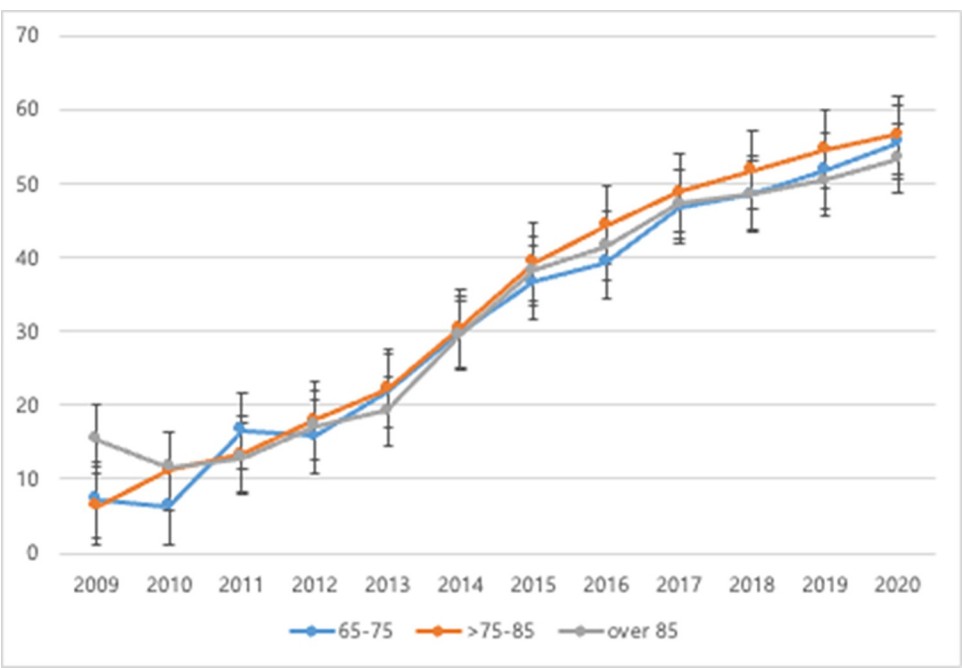

**Fig 4. Trends in inpatient CR participation rate by age group among patients aged ≥65.**

studies showed a median AHF hospitalization LOS of 7 to 10 days (5), and to the US, where a recent real-world study reported a mean AHF hospitalization LOS of 7 days [31] and 4 days [32]. The substantially longer LOS for Japanese patients may provide opportunities for more comprehensive and integrated treatment for this population. In addition, the Japanese health insurance systems provide reimbursement for rehabilitation during the hospitalization, including for acute care hospitals, which would contribute to longer LOS [33]. Our study also aligns with the previous study of Kanaoka and colleagues' findings. They determined that age, gender, body mass index, Barthel index, Charlson comorbidity index, and institutional

**Table 3. Characteristics of inpatient CR overall and by sex and age.**

| | Overall (N = 37,708) | Male (N = 19,205) | Female (N = 18,503) | Younger (<65) (N = 2,369) | Older (≥65) (N = 35,339) |
|---|---|---|---|---|---|
| **Length of stay (LOS)** | | | | | |
| | 26.2 (22.8) | 25.5 (22.7) | 27.0 (23.0) | 24.9 (22.8) | 26.3 (22.8) |
| **Time to initiation of CR program** | | | | | |
| | 4.6 (5.5) | 4.7 (5.6) | 4.6 (5.4) | 5.3 (7.2) | 4.6 (5.4) |
| **Number of CR sessions (1 session = 20 min)** | | | | | |
| | 15.5 (17.2) | 14.7 (16.6) | 16.2 (17.8) | 14.6 (16.8) | 15.5 (17.2) |
| **Total number of CR days*** | | | | | |
| | 12.4 (11.6) | 11.7 (11.0) | 13.0 (12.1) | 11.3 (12.2) | 12.4 (11.6) |
| **Days from CR initiation to completion** | | | | | |
| | 18.6 (18.8) | 17.8 (18.2) | 19.3 (19.3) | 17.2 (19.4) | 18.7 (18.7) |
| **Sessions per day during hospital stay** | | | | | |
| | 1.2 (0.5) | 1.2 (0.5) | 1.2 (0.5) | 1.3 (0.6) | 1.2 (0.5) |

Data are shown as mean (standard deviation).

*Total number of days that the patient participated in CR.

capacity were recognized as notable factors associated with participation in cardiac rehabilitation, both for inpatient and outpatient settings [34].

The observed increase in inpatient CR participation in Japan could be explained by hospital characteristics and increasing numbers of skilled healthcare professionals. Specifically, the trend observed in our study may reflect an increasing number of hospitals meeting the criteria for staffing and infrastructure required to be CR-certified hospitals. The Japanese Association of Cardiac Rehabilitation established a certification program for registered instructors of CR in 2000, and the Japanese Nursing Association launched a certification program in chronic HF nursing in 2012 [35]. Along with evolving hospital policies, certified healthcare professionals may influence trends in provision of CR, and hospitals not offering CR programs represent a diminishing proportion of hospitals overall.

The rate of CR participation has been persistently low for HF patients globally. In a study of the Get With The Guidelines-Heart Failure registry involving more than 100,000 patients at 338 institutions in the US, the rate of participation in outpatient CR was 2.3% among veterans and 2.6% among Medicare beneficiaries hospitalized for HF [36]. Analysis of data from the EUROASPIRE III survey showed underuse of CR overall in Europe, but with drastic variation between countries. The proportion of patients advised to follow a CR program ranged from 0.6% to 90.3%, and attendance rates of those advised to participate in CR varied from 0% to 94.9% [37]. Of those countries, proportion of patients advised to follow a cardiac rehabilitation program ranged between 0.6%, 0.8%, 4.9%, 7.3%, and 8.2% in Spain, Greece, Cyprus, Turkey, and Russian Federation, respectively. The attendance rates of those advised were between 0% to 45.8% among those 5 countries [37]. Our data show an outpatient CR rate that is consistent with the global trend.

Similar to our study, previous studies also reported low rates of outpatient CR participation among patients admitted for acute myocardial infarction [19,20], cardiovascular disease including acute myocardial infarction, angina pectoris, HF, peripheral artery disease, and post-cardiovascular surgery [21], and coronary heart disease [22]. Stable but low rates of outpatient CR participation were consistently seen among subgroups of AHF patients in our study. The low uptake of outpatient CR may be influenced by various patient and provider characteristics as well as system-level factors [38]. Patients who are older, those with severe disease states, high comorbidity burden, or low socioeconomic status may be less likely to participate in CR; lack of interest, fear of injury, and conflicting priorities may also act as barriers to uptake of CR [39]. Provider and the healthcare system-level factors including lack of resources and physicians, or physical therapists trained to deliver outpatient CR, as well as financial constraints and distance, have also been shown to correlate to a low rate of CR implementation [35,38–42].

There are currently 789 certified healthcare facilities offering inpatient CR in Japan, whereas only 690 certified healthcare facilities offer outpatient CR [43]. There was a moderate correlation observed between the number of facilities offering inpatient cardiac rehabilitation (r = 0.69), the count of JCS-certified cardiologists (r = 0.50), and the number of RICR (physicians [r = 0.52] and paramedical staff [r = 0.59]) with the count of patients who participated in inpatient cardiac rehabilitation [34]. This suggests that expanding the number of facilities is a significant factor in boosting participation in inpatient cardiac rehabilitation, potentially enhancing the quality of care for cardiovascular disease (CVD) patients admitted to these facilities. However, it's worth noting that the rate of inpatient cardiac rehabilitation participation exhibited substantial variation among different hospitals and was not uniform. Consequently, individual hospitals must undertake specific efforts to enhance the quality of inpatient care further.

In our current study, despite a significant rise in the number of hospitals offering CR, the outpatient CR participation rate remained exceptionally low, indicating that the state of outpatient CR participation in Japan has not seen improvement. This also suggests that factors beyond just an increase in CR facilities have influenced outpatient CR participation. These factors might encompass patient clinical characteristics [44–47], patient perceptions [45,46] and physician awareness regarding the benefits of CR [44–47].

The persistently low rate of outpatient CR participation was further underscored by the limited transition from inpatient CR to outpatient CR, implying that a majority of inpatient CR participants discontinued the program after their hospital discharge. This discontinuation could be attributed to issues like scheduling conflicts, transportation constraints, and the need to return to work sooner. To promote secondary prevention care effectively, an approach addressing these issues is imperative. In this regard, strategies that extend beyond a hospital-based program may be necessary. For instance, a home-based CR program could serve as a potential solution, as it addresses the challenge of hospital visits and appears to be equally effective when compared to center-based programs [48]. Moreover, advancements in telecommunications technology have made it possible to remotely monitor exercise and manage health [49,50]. These emerging technologies hold the promise of enhancing program accessibility and personalization [51], which is likely to boost participation in outpatient CR.

Studying the trends in CR rates in Japan has multifaceted implications and advantages for healthcare and policy considerations. It informs healthcare planning by revealing demand for cardiac rehabilitation, aids in resource allocation, and helps address gaps in access. The insights gained can lead to improved cardiac care and better overall health outcomes for patients.

## Study limitations

While the MDV includes data on more than 35 million patients, not all hospitals in Japan are in the MDV system, a sampling bias is thus possible, though hospital decisions to contribute their data are most likely to be driven by administrative reasons rather than patient or geographic characteristics, and we expect any bias to be non-differential with respect to our study variables. In addition, the MDV database does not include information on provider identifiers or characteristics, thus we were not able to describe variation in use of CR by providers or geographic region. In the MDV, health care services occurred outside of the MDV hospitals are not captured, which may have led to incomplete ascertainment of outpatient CR services. However, we required all patients in the cohort to have at least one encounter during one-year period prior to the AHF admission and during 6 months after discharge from AHF to ensure their active engagement in the healthcare system in MDV data and our results are in line with previous studies reporting the rate of outpatient CR in Japan.

## Conclusions

We found that over 40% of AHF patients in Japan participated in an intensive inpatient CR program. The rate of inpatient CR participation rose more than 6-fold over a 12-year period, whereas outpatient CR was vastly underutilized in post-AHF settings, with participation remaining stable but low (< 2%). Given the aging population and expected increases in the disease burden of AHF, it will be necessary to expand provision of multidisciplinary care including CR for HF and create infrastructure and incentives for patients and clinicians to increase the use of outpatient CR. Further studies are needed to understand the effectiveness of intensive inpatient CR programs for improving outcomes of HF patients.

## Supporting information

**S1 File. Supplemental material.**
(DOCX)

## Author Contributions

**Conceptualization:** Junghyun Kim, Soko Setoguchi.

**Formal analysis:** Jenny Jiang.

**Investigation:** Junghyun Kim.

**Methodology:** Junghyun Kim.

**Project administration:** Junghyun Kim.

**Resources:** Jenny Jiang, Sophie Shen.

**Software:** Sophie Shen.

**Supervision:** Soko Setoguchi.

**Visualization:** Junghyun Kim.

**Writing – original draft:** Junghyun Kim.

**Writing – review & editing:** Sophie Shen, Soko Setoguchi.

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
