## [Decision Letter · Decision Letter 0]

9 Aug 2023

PONE-D-23-12827

Trends in Cardiac Rehabilitation Rates among Patients admitted for Acute Heart Failure in Japan, 2009-2020

PLOS ONE

Dear Dr. Kim,

Thank you for submitting your manuscript to PLOS ONE. After careful consideration, we feel that it has merit but does not fully meet PLOS ONE’s publication criteria as it currently stands. Therefore, we invite you to submit a revised version of the manuscript that addresses the points raised during the review process.

Please refer to the comments by the two reviewers below and make your amendments accordingly. 

We look forward to receiving your revised manuscript.

Kind regards,

Shairyzah Ahmad Hisham, PhD.

Academic Editor

PLOS ONE

Journal Requirements:

2. Thank you for stating the following in the Competing Interests/Financial Disclosure * (delete as necessary) section:

"Funding: The study was funded by Bristol Myers Squibb, NJ, USA (CV013-030).

Disclosures: Soko Setoguchi and Junghyun Kim received consultancy fees from Bristol Myers Squibb. Jenny Jiang and Sophie Shen are employees of Bristol Myers Squibb, and Junghyun Kim was an employee of Rutgers University when the research was conducted. "

We note that you received funding from a commercial source: "Bristol-Myers Squibb"

"Disclosures: Soko Setoguchi and Junghyun Kim received consultancy fees from Bristol Myers Squibb. Jenny Jiang and Sophie Shen are employees of Bristol Myers Squibb, and Junghyun Kim was an employee of Rutgers University when the research was conducted."

4.In your Data Availability statement, you have not specified where the minimal data set underlying the results described in your manuscript can be found. PLOS defines a study's minimal data set as the underlying data used to reach the conclusions drawn in the manuscript and any additional data required to replicate the reported study findings in their entirety. All PLOS journals require that the minimal data set be made fully available. For more information about our data policy, please see http://journals.plos.org/plosone/s/data-availability.

**Comments to the Author**

1. Is the manuscript technically sound, and do the data support the conclusions?

Reviewer #1: Yes

Reviewer #2: Yes

2. Has the statistical analysis been performed appropriately and rigorously? 

Reviewer #1: Yes

Reviewer #2: Yes

3. Have the authors made all data underlying the findings in their manuscript fully available?

Reviewer #1: Yes

Reviewer #2: No

4. Is the manuscript presented in an intelligible fashion and written in standard English?

Reviewer #1: Yes

Reviewer #2: Yes

5. Review Comments to the Author

Reviewer #1: The manuscript has been prepared fairly well, but there are a number of things that need clarification or refinement.

1. Please add more specific what the implication of the study and what the benefit of knowing the trend of CR rates in Japan.

2. Many variations in patient characteristics were observed in this study, including the Charlson comorbidity index, but no discussion related to this factor. Is there any influence of comorbidities or Charlson Index on the participation as well as the duration of CR?

3. Table 3 presents the characteristics of inpatient CR that are only differentiated by gender and age. How about other relevant factors that affect the characteristics of the inpatient CR?

4. It is interesting that inpatient CR has increased significantly, but the outpatient CR rate remains low. The number of certified healthcare facilities offer outpatient CR is not so big different with that offering inpatient CR in Japan (690 vs 789), but the rate of participation is not proportional. Please discuss more dept about the possible factor in Japan.

Reviewer #2: 1. Please include the detail approval number from the institutional review board at Rutgers University.

2. Explain whether exemption for consent was obtained from the ethical committee since this is a retrospective study.

3. To clarify on this statement " lack of resources and physicians or physical therapists trained deliver outpatient CR", do you mean trained to deliver outpatient CR?

4. Please include another paragraph on suggestions of how to improve CR rate in outpatient settings.

---

## [Author Response · Author response to Decision Letter 0]

10 Oct 2023

Reviewer #1: The manuscript has been prepared fairly well, but there are a number of things that need clarification or refinement.

1. Please add more specific what the implication of the study and what the benefit of knowing the trend of CR rates in Japan.

Thank you very much for your comments, I have added the implications and the benefit of addressing the trend of CR rates in Japan as follow and highlighted in yellow in the manuscript: 

Studying the trends in Cardiac Rehabilitation (CR) rates in Japan has multifaceted implications and advantages for healthcare and policy considerations. It informs healthcare planning by revealing demand for cardiac rehabilitation, aids in resource allocation, and helps address gaps in access. The insights gained can lead to improved cardiac care and better overall health outcomes for patients.

2. Many variations in patient characteristics were observed in this study, including the Charlson comorbidity index, but no discussion related to this factor. Is there any influence of comorbidities or Charlson Index on the participation as well as the duration of CR?

In the Baseline characteristics of AHF patients by CR participation in Result section, there were mentioned regarding comorbidities as Compared to patients who did not receive CR, those who participated in any CR had higher rates of several comorbidities including coronary artery disease, hypertension, AF, and hyperlipidemia. However, the prevalence of COPD and chronic renal disease were slightly higher in patients with no CR compared to those with any CR. 

In addition, I have added a previous study that also indicate the comorbidity index had an impact on CR as follows: 

Our study also aligns with the previous study of Kanaoka and colleagues’ findings. They determined that age, gender, body mass index, Barthel index, Charlson comorbidity index, and institutional capacity were recognized as notable factors associated with participation in cardiac rehabilitation, both for inpatient and outpatient settings [34].

3. Table 3 presents the characteristics of inpatient CR that are only differentiated by gender and age. How about other relevant factors that affect the characteristics of the inpatient CR? 

Thank you very much for your valued comments.

In Table 2 describes other characteristics of inpatient CR, supplementary Figure B to F depicts of inpatient CR by NYHA class, BMI, smoking status, diabetes status, and AF status. Participation in inpatient CR declined over the study period among NYHA class 1 patients but rose among all other NYHA classifications (Supplementary Figure B). Patients with diabetes participated in inpatient CR slightly more frequently than those without diabetes throughout the study period (Supplementary Figure E), and patients with AF participated in inpatient CR more often than those without AF beginning in 2011 (Supplementary Figure F). 

In addition, other relevant factors from supported papers also added in the manuscript as follows:

There was a moderate correlation observed between the number of facilities offering inpatient cardiac rehabilitation (r=0.69), the count of JCS-certified cardiologists (r=0.50), and the number of RICR (physicians [r=0.52] and paramedical staff [r=0.59]) with the count of patients who participated in inpatient cardiac rehabilitation [34]. In our current study, despite a significant rise in the number of hospitals offering cardiac rehabilitation (CR), the outpatient cardiac rehabilitation (OCR) participation rate remained exceptionally low, indicating that the state of OCR participation in Japan has not seen improvement. This also suggests that factors beyond just an increase in CR facilities have influenced OCR participation. These factors might encompass patient clinical characteristics [44-47], patient perceptions [45, 46] and physician awareness regarding the benefits of CR [44-47]. 

The persistently low rate of OCR participation was further underscored by the limited transition from inpatient cardiac rehabilitation (ICR) to OCR, implying that a majority of ICR participants discontinued the program after their hospital discharge. This discontinuation could be attributed to issues like scheduling conflicts, transportation constraints, and the need to return to work sooner.

4. It is interesting that inpatient CR has increased significantly, but the outpatient CR rate remains low. The number of certified healthcare facilities offer outpatient CR is not so big different with that offering inpatient CR in Japan (690 vs 789), but the rate of participation is not proportional. Please discuss more dept about the possible factor in Japan.

Really appreciate your comments and as you suggested other possible factors in Japan regarding CR rates are addressed in the manuscript and as follows:

There was a moderate correlation observed between the number of facilities offering inpatient cardiac rehabilitation (r=0.69), the count of JCS-certified cardiologists (r=0.50), and the number of RICR (physicians [r=0.52] and paramedical staff [r=0.59]) with the count of patients who participated in inpatient cardiac rehabilitation [34]. This suggests that expanding the number of facilities is a significant factor in boosting participation in inpatient cardiac rehabilitation, potentially enhancing the quality of care for cardiovascular disease (CVD) patients admitted to these facilities. However, it's worth noting that the rate of inpatient cardiac rehabilitation participation exhibited substantial variation among different hospitals and was not uniform. Consequently, individual hospitals must undertake specific efforts to enhance the quality of inpatient care further.

Reviewer #2: 

1. Please include the detail approval number from the institutional review board at Rutgers University.

Thank you very much for informing that we need to add the specific information. The IRB approval number (Pro2019002578) added in the manuscript. 

2. Explain whether exemption for consent was obtained from the ethical committee since this is a retrospective study.

Yes, the phrase “exemption from ethics approval and waiver of informed consent granted” has added in the sentence. 

3. To clarify on this statement " lack of resources and physicians or physical therapists trained deliver outpatient CR", do you mean trained to deliver outpatient CR?

Yes, it was meant by “trained to deliver outpatient CR” I added “to” into the sentence, thank you for checking the details.

4. Please include another paragraph on suggestions of how to improve CR rate in outpatient settings.

As you requested with valuable thought, more suggestions of how to improve outpatient CR rates are addressed the below:

The persistently low rate of outpatient CR participation was further underscored by the limited transition from inpatient CR to outpatient CR, implying that a majority of inpatient CR participants discontinued the program after their hospital discharge. This discontinuation could be attributed to issues like scheduling conflicts, transportation constraints, and the need to return to work sooner. To promote secondary prevention care effectively, an approach addressing these issues is imperative. In this regard, strategies that extend beyond a hospital-based program may be necessary. For instance, a home-based CR program could serve as a potential solution, as it addresses the challenge of hospital visits and appears to be equally effective when compared to center-based programs [48]. Moreover, advancements in telecommunications technology have made it possible to remotely monitor exercise and manage health [49, 50]. These emerging technologies hold the promise of enhancing program accessibility and personalization [51], which is likely to boost participation in outpatient CR.

---

## [Editor Report · Decision Letter 1]

10 Nov 2023

Trends in Cardiac Rehabilitation Rates among Patients admitted for Acute Heart Failure in Japan, 2009-2020

PONE-D-23-12827R1

Dear Dr. Kim,

We’re pleased to inform you that your manuscript has been judged scientifically suitable for publication and will be formally accepted for publication once it meets all outstanding technical requirements.

Kind regards,

Shairyzah Ahmad Hisham, PhD.

Academic Editor

PLOS ONE

Additional Editor Comments (optional):

The authors have addressed all comments and issues brought forward by the 2 reviewers and the manuscript is now of acceptable standard for publication. Congratulations!

---

## [Editor Report · Acceptance letter]

16 Nov 2023

PONE-D-23-12827R1 

Trends in Cardiac Rehabilitation Rates among Patients admitted for Acute Heart Failure in Japan, 2009-2020 

Dear Dr. Kim:

I'm pleased to inform you that your manuscript has been deemed suitable for publication in PLOS ONE. Congratulations! Your manuscript is now with our production department. 

Kind regards, 

on behalf of

Dr. Shairyzah Ahmad Hisham 

Academic Editor

PLOS ONE